# Protein Kinase C (PKC) Isozymes as Diagnostic and Prognostic Biomarkers and Therapeutic Targets for Cancer

**DOI:** 10.3390/cancers14215425

**Published:** 2022-11-03

**Authors:** Takahito Kawano, Junichi Inokuchi, Masatoshi Eto, Masaharu Murata, Jeong-Hun Kang

**Affiliations:** 1Center for Advanced Medical Innovation, Kyushu University, 3-1-1 Maidashi, Higashi-ku, Fukuoka 812-8582, Japan; 2Department of Urology, Graduate School of Medical Sciences, Kyushu University, Maidashi, Higashi-ku, Fukuoka 812-8582, Japan; 3Division of Biopharmaceutics and Pharmacokinetics, National Cerebral and Cardiovascular Center Research Institute, 6-1 Shinmachi, Kishibe, Suita, Osaka 564-8565, Japan

**Keywords:** protein kinase C, biomarker, diagnosis, prognosis, therapeutic target, poor survival, cancer treatment

## Abstract

**Simple Summary:**

Protein kinase C (PKC) isozymes play key roles in the proliferation, differentiation, survival, migration, invasion, apoptosis, and anticancer drug resistance of cancer cells. PKC isozymes are attractive therapeutic targets for cancer and have great potential as diagnostic and prognostic biomarkers for diagnosing cancers and for predicting disease-free survival and survival rates, respectively. This review discusses the potential of PKC isozymes as diagnostic and prognostic biomarkers and therapeutic targets for cancer.

**Abstract:**

Protein kinase C (PKC) is a large family of calcium- and phospholipid-dependent serine/threonine kinases that consists of at least 11 isozymes. Based on their structural characteristics and mode of activation, the PKC family is classified into three subfamilies: conventional or classic (cPKCs; α, βI, βII, and γ), novel or non-classic (nPKCs; δ, ε, η, and θ), and atypical (aPKCs; ζ, ι, and λ) (PKCλ is the mouse homolog of PKCι) PKC isozymes. PKC isozymes play important roles in proliferation, differentiation, survival, migration, invasion, apoptosis, and anticancer drug resistance in cancer cells. Several studies have shown a positive relationship between PKC isozymes and poor disease-free survival, poor survival following anticancer drug treatment, and increased recurrence. Furthermore, a higher level of PKC activation has been reported in cancer tissues compared to that in normal tissues. These data suggest that PKC isozymes represent potential diagnostic and prognostic biomarkers and therapeutic targets for cancer. This review summarizes the current knowledge and discusses the potential of PKC isozymes as biomarkers in the diagnosis, prognosis, and treatment of cancers.

## 1. Introduction

Protein kinase-mediated phosphorylation of serine (S), threonine (T), and/or tyrosine (Y) residues in target proteins is involved in the activation or inactivation of intracellular signal transduction pathways. Protein kinase C (PKC) is a family of calcium- and phospholipid-dependent serine/threonine kinases. The PKC family consists of at least 11 isozymes and is classified into three subfamilies based on their structural characteristics and mode of activation: conventional or classic (cPKCs; α, βI, βII, and γ), novel or non-classic (nPKCs; δ, ε, η, and θ), and atypical (aPKCs; ζ, ι, and λ) (PKCλ is the mouse homolog of PKCι) PKC isozymes [1,2].

All PKCs consist of a regulatory and catalytic (kinase) domain. The regulatory region is divided into an autoinhibitory domain (pseudosubstrate) and two membrane-targeting domains (C1 and C2). The C1 and C2 domains bind to diacylglycerol (DAG) and Ca^2+^, respectively. The C3 and C4 domains in the catalytic region bind to ATP and its target substrate, respectively. The C1 domain mediates DAG-dependent translocation of cPKCs and nPKCs, but not of aPKCs, which contain a single C1 domain. cPKCs contain the calcium-sensitive C2 domain and bind to Ca^2+^, whereas nPKCs (contain an atypical C2-like domain) and aPKCs (without the C2 domain) do not. A phosphatidylserine (PS)-binding domain is not found in all PKCs, but PS, either alone or in combination with DAG and Ca^2+^, is essential for the phosphorylation of the target substrate [1,2]. The consensus phosphorylation site motifs for PKCs are (R/K)X(S/T), (R/K)(R/K)X(S/T), (R/K)XX(S/T), (R/K)X(S/T)XR/K, and (R/K)XX(S/T)XR/K, which clearly show that PKC substrates are typically rich in basic amino acids (arginine (R) and/or lysine (K)) [3]. PKC isozyme-specific substrates and their design methods have been extensively reviewed in previous articles [3,4,5].

PKC isozymes play key roles in the proliferation, differentiation, survival, migration, invasion, apoptosis, and anticancer drug resistance of cancer cells. Because of their high potential as therapeutic targets, many natural and synthetic PKC inhibitors have been developed and tested in clinical trials for cancer treatment (for review, see [6,7]). Furthermore, PKC isozymes also have great potential as diagnostic and prognostic biomarkers for diagnosing cancers and for predicting disease-free survival and survival rates (Figure 1). This review discusses the potential of PKC isozymes as diagnostic and prognostic biomarkers and therapeutic targets for cancer.

## 2. PKC Isozymes as Prognostic Biomarkers or Therapeutic Targets for Cancer

### 2.1. Bladder Cancer

Among the PKC isozymes, PKCα, βI, βII, δ, ε, η, and ζ have been observed in bladder cancer cells and tissues. PKCβI, βII, δ, and η are found mainly in early-stage bladder cancer, but their levels are reduced as cancer progresses. PKCα and ζ levels increase with increasing cancer stage [8,9,10].

In a large-scale multi-omics analysis, elevated expression of PKCα protein was associated with poor prognosis in patients with bladder cancer, in addition to increased expression of beclin, epidermal growth factor receptor (EGFR), annexin-1, and AXL proteins and downregulation of Src protein [11]. A previous study demonstrated that PKCα/β has a critical role in phospholipase Cε-mediated bladder cancer cell invasion and migration [12], and cell proliferation [13]. Furthermore, the expression of PKCα and nuclear factor kappa-B (NF-κB) in bladder cancer cells positively correlated with poor prognosis [14]. PKCα induced cellular resistance to apoptosis by stimulating NF-κB activation [14,15].

High PKCα activity, high netrin-1 expression, and low UNC5B expression enhanced the tolerance of bladder cancer cells to cisplatin, whereas the opposite expression pattern increased their sensitivity to cisplatin treatment [16]. Overexpression of tripartite motif 29 (TRIM29) upregulated the levels of cell survival-related proteins (e.g., cyclin and Bcl family) and inhibited cisplatin-mediated cell apoptosis in bladder cancer cells. However, its expression was downregulated following treatment with the PKC inhibitor staurosporine or the NF-κB inhibitor BAY 11-7082. These results indicate that TRIM29 inhibits drug-induced apoptosis in bladder cancer via the PKC/NF-κB signaling pathway [17]. Moreover, in patients treated with the anticancer drug adriamycin, high PKCα level is associated with a shorter recurrence-free period and higher drug resistance than low PKCα level [18]. However, PKCα inhibition induces apoptosis in bladder cancer cells by enhancing the activities of caspase-3 and poly (ADP-ribose) polymerase (PARP) [19]. These studies suggest that PKCα activity in bladder cancer may be a biomarker for poor prognosis and anticancer drug resistance and that PKCα inhibition may be a useful therapeutic option for bladder cancer.

In contrast, loss of aPKC (PKCι and ζ) expression in superficial bladder cancer is associated with a high recurrence rate and poor survival [20]. Treatment with the aPKC inhibitors ζ-Stat and 5-amino-1-2,3-dihydroxy-4-(methylcyclopentyl)-1H-imidazole-4-carboxamide (ICA-1), together with rapamycin, blocked bladder cancer progression [21].

### 2.2. Blood and Bone Marrow Cancers

Blood and bone marrow cancers can be divided into three major types: multiple myeloma (MM), leukemia, and lymphoma.

#### 2.2.1. MM

MM is a type of bone marrow cancer. Very few studies have examined the role or function of PKC isozymes in MM. PKCβ has attracted immense attention as a therapeutic target in MM [22,23]; however, in clinical trials, treatment with the oral inhibitor enzastaurin showed no clinical benefit in patients with MM [24].

#### 2.2.2. Leukemia

Based on the cell of origin, leukemia is classified as lymphocytic (lymphoid or lymphoblastic) or myeloid (myelogenous or myeloblastic) types and further divided into four types: acute lymphocytic leukemia (ALL), chronic lymphocytic leukemia (CLL), acute myeloid leukemia (AML), and chronic myeloid leukemia (CML).

##### ALL and CLL

Expression of PKCβ, γ, δ, and ζ was found in all patients with CLL, and that of PKCα, ε, and ι was variable, whereas PKCθ was not expressed [25]. Activated PKCα/βII (Thr638/641) was higher in patients with differentiated B-cell CLL compared to that in healthy controls [26]. The interaction of Rack1 and PKCα, but not PKCβ, was observed in two T-cell ALL-derived cell lines (Jurkat and CCRF-CEM). PKCα inhibition increased apoptosis in Rack1-overexpressing T-cell ALL cells following treatment with chemotherapeutic drugs [27]. In ALL, overexpression of PKCα did not affect cell proliferation, cell cycle, or activation of mitogen-activated protein kinases (MAPKs), but increased chemoresistance through Bcl-2 activation [28]. These studies suggest that PKCα may be closely associated with increased chemoresistance in lymphocytic leukemia.

Among the PKC isozymes, PKCβ is considered to be a useful therapeutic target for lymphocytic leukemia as it participates in cell survival and proliferation [29,30,31], resistance to apoptosis [32], and chemoresistance induced by stromal cells, which are key components of the lymphocytic leukemia microenvironment [30,33]. However, PKCβ-specific inhibitors have failed to show significant clinical benefits in patients with lymphocytic leukemia [7].

In addition, PKCε [34] and PKCδ have been reported to mediate leukemic cell survival [35] and cell sensitivity to anticancer drugs induced by PKCζ overexpression [36]. Furthermore, links between PKCδ and Notch2 [37] and PKCθ and Notch1 signaling in leukemic cells [38] and aPKCλ/ι-mediated transformation of B-cell progenitors (can generate B-cell ALL) by BCR-ABL [39] have been reported.

##### AML and CML

PKCα activation is associated with poor survival in patients with AML [40]. PKCα activation also enhanced resistance to chemotherapy in AML cells through Bcl-2 phosphorylation [41] and extracellular-signal-regulated kinase 1/2 (ERK1/2) and Akt activation [42]. PKCα inhibition enhanced selenite-induced apoptosis of the acute promyelocytic leukemia cell line NB4 [42].

PKCε was found to be markedly overexpressed in patients with AML and positively correlated with reduced complete remission, disease-free survival, and enhanced resistance to the chemotherapeutic agent daunorubicin through P-glycoprotein (P-gp)-mediated drug efflux [43]. PKCε overexpression protects AML cells from mitochondrial reactive oxygen species (ROS)-inducing agents. However, PKCε deletion reduced patient-derived AML cell survival and disease onset in an AML mouse model [44].

There was a significant association between reduced PKCδ levels and relapse in patients with AML [45]. PKCδ appears to be involved in stimulating anticancer drug-mediated apoptosis through caspase-3 activation [46,47], phosphorylation of eukaryotic initiation factor-α [45], and downregulation of heterogeneous nuclear ribonucleoprotein K [48].

A recent phase III trial showed that treatment with midostaurin (also known as PKC412; CGP 41251) with standard chemotherapy significantly prolonged overall and event-free survival in patients with mutant *FLT3*-positive AML [49]. Although midostaurin was originally developed as a PKC inhibitor, its clinical benefits are mainly achieved via tyrosine kinase inhibition [50]. Midostaurin has been approved by the FDA for the treatment of newly diagnosed adult patients with mutant *FLT3*-positive AML and adult patients with systemic mastocytosis with associated hematological neoplasm or mast cell leukemia, which is an aggressive subtype of AML [7].

The addition of the PKC inhibitor staurosporine increases the sensitivity of imatinib-resistant CML to imatinib by inducing G2/M phase arrest through PKCα-dependent CDC23 downregulation [51]. PKCβ overexpression in CML cells also enhances resistance to imatinib through arachidonate 5-lipoxygenase (Alox5) signaling. Alox5 levels were increased in both bone marrow biopsies and CD34^+^ cells derived from patients with imatinib-resistant CML. In contrast, prolonged survival was observed in CML mice treated with imatinib in combination with the PKCβ inhibitor LY333531 [52]. PKCη was upregulated in samples from patients with CML with BCR-ABL-independent imatinib resistance or CML stem cells, leading to sustained RAF/MEK/ERK signaling following imatinib treatment. Combined treatment with imatinib and the MEK inhibitor trametinib prolonged survival in mouse models of BCR-ABL-independent imatinib-resistant CML [53]. In addition, aPKCλ/ι may be a potential therapeutic target for treating tyrosine kinase inhibitor (TKI)-resistant CML [39].

##### Myelodysplastic Syndromes (MDSs)

MDSs are a heterogeneous group of hematopoietic stem cell disorders and frequently evolve into AML [54,55]. Nuclear translocation of PKCα induced erythropoiesis in patients with low-risk MDS following treatment with the immunomodulatory drug lenalidomide [56]. Furthermore, the PI-PLCβ1/cyclin D3/PKCα signaling pathway was associated with iron-induced oxidative stress and ROS production in MDS patients [57].

#### 2.2.3. Lymphoma

Lymphoma begins in the T or B cells of the lymphatic system and is classified into two major subtypes: Hodgkin and non-Hodgkin lymphoma (NHL). PKC isozyme analysis using reactive lymphoid tissues, human B-cell lymphoma, and human lymphoma cell lines revealed that PKCα, βII, γ, and δ were expressed in B-cell malignancies. Compared to other types of lymphomas, Burkitt’s lymphomas overexpress PKCα. In Burkitt’s lymphoma, the overall survival was higher in PKCγ-positive cases than in PKCγ-negative cases [58]. PKCζ, but not cPKC, is involved in the regulation of telomerase activity in Burkitt’s lymphoma cells [59].

In follicular lymphomas, PKCβII is overexpressed, mainly in the mantle and marginal zones. PKCβII expression was also found in most angioimmunoblastic T-cell lymphomas, lymphoblastic T-cell lymphomas, and marginal zone/mucosa-associated lymphoid tissue lymphomas, although the pattern of expression was very heterogeneous. However, PKCβII expression was not observed in Hodgkin’s disease or anaplastic large-cell lymphoma [60]. Higher PKCβII expression was noted in human immunodeficiency virus-infected patients than in uninfected patients with diffuse large B-cell lymphoma (DLBCL), which is the most common subtype of NHL [61]. In DLBCL, higher PKCβ expression was found in the activated B-cell-like subtype than in the germinal center B-cell-like subtype, and its elevated levels were associated with worse survival in both subtypes [62]. PKCβII expression in DLBCL was correlated with poor overall and progression-free survival in patients treated with cyclophosphamide, doxorubicin (DOX), vincristine, and prednisolone [63]. PKCβII expression was associated with worse 5-year event-free and overall survival in patients with nodal DLBCL, especially in patients with low-risk International Prognostic Index [64,65,66]. Based on these reports, PKCβII is regarded as a marker for poor prognosis and a chemotherapeutic target for lymphoid malignancies.

In lymphoma, PKCδ activation stimulates anticancer drug-mediated apoptosis through caspase-3 activation [67,68], JNK activation [69], or phosphorylation and activation of lysosomal acidic sphingomyelinase [70]. The PKCζ/mammalian target of rapamycin (mTOR) pathway may also be a therapeutic target for rituximab-mediated treatment of follicular lymphoma [71].

### 2.3. Brain Cancer (Glioblastoma)

Glioblastoma is a high-grade astrocytoma and the most malignant type of brain tumor. Astrocytoma malignancies are positively correlated with progesterone receptor (PR) and PKCα levels as well as with the intracellular colocalization of these proteins. Patients with astrocytoma grades III and IV with low expression of *PGR* and *PRKCA* mRNA showed higher survival than those with high expression [72]. Treatment with mTOR inhibitors (rapamycin, temsirolimus, torin-1, and PP242) reduces glioblastoma progression by reducing invasion, migration, and matrix metalloproteinase (MMP) activity (MMP2 and MMP9) through the reduction of PKCα and NF-κB signaling pathways [73]. Furthermore, PKCα/phosphoinositide 3-kinase (PI3K) signaling pathways increase astrocytoma invasion by downregulating low-density lipoprotein receptor-related protein [74]. Overexpression of the long noncoding RNA TCONS_00020456, which targets the Smad2/PKCα axis, reduced glioma cell proliferation, migration, and invasion and inhibited epithelial–mesenchymal transformation and glioma progression in vivo [75]. Activation of the lysophosphatidic acid receptor LPA_1_ induces PKCα translocation to the nucleus, inhibits the LPA_1_/PKCα axis, and reduces glioblastoma growth and progression [76,77].

Although PKCα is a therapeutic target for glioblastoma, a previous study showed no clinical benefits in patients with high-grade gliomas following treatment with the antisense oligonucleotide aprinocarsen directed against PKCα [78]. However, a recent study suggested that combination therapy with JAK2 (AZD1480) and a PKCα inhibitor (erlotinib or osimertinib) induced apoptosis of glioblastoma, which is the most malignant and aggressive form of astrocytoma, in both flank and in patient-derived orthotopic xenograft models, indicating that PKCα and JAK2 may be therapeutic targets for glioblastoma [79]. Interestingly, in vitro experiments using U87MG cells showed that loss of PKCα proteins inhibited cell growth or survival, but the same effects were not obtained by inhibiting PKCα activity, indicating that ATP-competitive inhibitors of PKCα may have little or no therapeutic effect in glioblastoma [80].

PKCι is associated with cell proliferation [81,82,83], survival [84], invasion [81,83], apoptosis [82], and anticancer resistance [85] in glioblastomas. PKCι is overexpressed and activated in patient-derived glioblastoma stem-like cells compared to normal neural stem cells and normal brain lysates [86]. Glioblastoma cell proliferation depends on the PI3K/PKCι/CDK7/CDK2 pathway [82], and cell survival depends on the PI3K/PDK1/PKCι/BAD pathway [87]. Moreover, elevated PKCι level increases resistance to cisplatin in glioblastoma cells by suppressing GMFβ/p38 MAPK signaling [85] and induces glioblastoma motility by coordinating the formation of a single leading-edge lamellipod [81]. These results demonstrated that PKCι may be an important therapeutic target for glioblastoma.

Elevated PKCι activation in glioblastoma cells increased their sensitivity to PKCι inhibitors, but low PKCι activation resulted in both Src activation and sensitivity to Src inhibitors. The combination of PKCι and Src inhibitors prolonged survival beyond that of either drug alone [84]. The combination of PKCι inhibitors ICA-1 and temozolomide also decreased the invasion of glioblastoma cell lines and reduced glioblastoma growth and volume in mice [83]. Furthermore, combined treatment with ICA-1 and tumor necrosis factor-related apoptosis-inducing ligand (TRAIL) stimulated caspase-3-mediated apoptosis in glioblastoma cells by downregulating PKCι and c-Jun [88].

As mentioned above, PKCα activation leads to increased proliferation and decreased apoptosis of glioblastoma cells. However, activated PKCδ has the opposite effect resulting in decreased proliferation and increased apoptosis [89,90,91] through Bcl-2 phosphorylation [91] or Akt (also called PKB) inhibition [90].

PKCε overexpression was found in primary pediatric anaplastic astrocytoma (grade III) tumor samples as well as in glioblastoma multiforme (grade IV) and gliosarcoma tumor samples, but not in pilocytic astrocytomas (grade I) [92]. PKCε inhibition decreased the expression of Beclin1, Atg5, and PI3K in glioma cells and increased the expression of the autophagy-related proteins mTOR and Bcl-2. PKCε knockdown also reduced the adhesion of glioblastoma cells by decreasing total focal adhesion kinase (FAK) protein levels and its phosphorylation [93].

In addition to PKCδ and ε as potential therapeutic targets for glioblastoma, PKCζ may be a therapeutic target for glioblastoma cell migration and invasion [94], and PKCη may be a therapeutic target for glioblastoma proliferation [95,96].

### 2.4. Breast Cancer

Several PKC isozymes, including PKCα, β, δ, ε, ζ, η, ζ, θ, and λ, have been identified in breast cancer. PKCα, βI, and βII levels in breast cancer specimens and PKCβII levels in HER2-positive cancers are higher than those in adjacent normal breast tissues [97]. A previous study reported enhanced PKCε levels in high histologic grade and human epidermal growth factor receptor-2 (HER2/ErbB2)-positive, estrogen receptor (ER)-negative, and PR-negative breast cancers [98], whereas another study suggested that PKCε is downregulated in all cancer stages, molecular subtypes, metastatic and nonmetastatic groups, and patients with or without anticancer drug treatment compared to healthy controls [99].

PKCα is closely associated with poor survival in patients with breast cancer and increased anticancer resistance. In fact, poorer survival was observed in patients with PKCα-positive breast cancer than in those with PKCα-negative breast cancer. PKCα levels are positively associated with estrogen and PR negativity, cancer grade, and proliferative activity [100]. In endocrine-resistant and triple-negative breast cancer cell lines, PKCα plays a key role in maintaining their migratory and invasive phenotypes through FOXC2-mediated repression of p120-catenin [101].

PKCα may also be used as a marker for estrogen resistance because of the positive association between high PKCα levels and enhanced resistance to antiestrogen hormonal therapy (e.g., tamoxifen) [102,103]. PKCα levels are positively correlated with triple-negative breast cancers that are characterized by a lack of ER, PR, or ErbB2 expression [104,105], and there is an inverse relationship between PKCα levels and ERα expression [106]. Furthermore, PKCα showed relatively higher basal activity in drug-resistant MCF-7/ADR cells than in drug-sensitive MCF-7 cells. Inhibition of PKCα activity improves intracellular accumulation of DOX in MCF-7/ADR cells [107].

Overexpression of the Notch1 receptor and its ligand Jagged-1 is associated with poor survival in patients with ErbB-positive breast cancer [108,109] and increased trastuzumab resistance [110]. However, activated PKCα attenuates Jagged-1-mediated Notch1 activity in ErbB2-positive breast cancer and restores trastuzumab resistance, suggesting that PKCα activity may be a potential prognostic marker for low Notch activity and increased trastuzumab sensitivity in ErbB2-positive breast cancer [111].

Combined treatment with a PKC inhibitor and all-trans-retinoic acid (ATRA) reduced the growth, self-renewal, and frequency of cancer stem cells (CSCs) in a retinoic acid receptor (RAR) signaling-dependent manner. Low PKCα and high RAR levels were associated with significantly increased relapse-free survival (RFS) in patients with ER-negative breast cancer [112]. However, another study reported that PKCα overexpression promotes RARα expression levels in breast cancer cells following ATRA treatment, and increased RARα leads to ATRA sensitization through AP1 trans-repression [113].

ErbB2 entry into the endocytic recycling compartment stimulated by PKCα and PKCδ [114], or PKCδ-mediated Src activation, promotes ErbB2-induced mammary tumorigenesis [115]. Furthermore, human breast CSCs efficiently formed tumor xenografts in nude mice; however, their tumorigenesis was markedly reduced by PKCδ inhibition. In the mesenchymal CSC-like MCF10C cell line (M3), which is derived from MCF10A (M1) cells, PKCδ inhibition blocked tumor spheroid formation [116]. These data indicate that PKCδ is associated with mammary tumorigenesis and may be a predictive marker.

In contrast, in breast cancer samples from patients, high PKCδ and PKCα expression was correlated with endocrine responsiveness and ER negativity, respectively. A longer duration of endocrine response is observed in patients with a PKCδ(+)/PKCα(−) than the PKCδ(+)/PKCα(+) phenotype, indicating that PKCδ may be useful for predicting the response to antiestrogen therapy [117]. Interestingly, AD198 (a DOX analog)-induced apoptosis is PKCδ-dependent [118], but PKCδ in normal murine mammary gland cells increased resistance against AD198-mediated cell death through Akt and NF-κB survival pathways [119].

The high PKCζ group exhibits poorer prognosis, including advanced clinical stage, more lymph node involvement, larger tumor size, and lower disease-free and overall survival rates, compared to the low PKCζ group [120]. Moreover, PKCζ levels are higher in invading tissues than in non-invading tissues and are more abundant in ductal tissues than in lobular tissues. Its invasive behavior is induced through the Ras-related C3 botulinum toxin substrate 1 (Rac1) and Ras homolog gene family member A (RhoA) pathways. These results suggest that PKCζ may be used as an indicator of bladder cancer invasion [121] and a prognostic marker for breast cancer.

PKCθ-induced phosphorylation of Fra-1 stimulates the migration of breast cancer cells, and phosphorylated Fra-1 expression is enriched at the invasive front of human breast cancer cells. Furthermore, PKCθ is positively associated with MMP1 mRNA expression in human breast cancer samples [122]. PKCθ is enriched in circulating tumor cells in patients with triple-negative breast cancer brain metastases. Nuclear PKCθ-positive phenotype, together with cell surface vimentin-positive and ABCB5-positive phenotypes, a CSC-like marker associated with therapeutic resistance, is found in a higher proportion in brain metastases of patients with breast cancer than in primary breast tumors, indicating an association between PKCθ and cancer metastasis [123]. Enhanced PKCθ levels in triple-negative breast cells activate growth factor-independent growth, anoikis resistance, and migration [124]. Therefore, PKCθ upregulation may be used as a marker for predicting migratory and invasive behaviors in breast cancer cells.

Pal’s group reported that PKCη may serve as a potential biomarker for breast cancer malignancy because of higher expression of PKCη in malignant cells than in non-tumorigenic or pre-malignant cells, and they also reported a positive correlation between PKCη levels and increased breast cancer cell growth or clonogenic survival [125]. Increased PKCη expression in post-chemotherapy biopsies of patients with advanced and aggressive breast cancers was correlated with poor survival, showing that PKCη may also be an indicator of poor survival and a predictor of the effectiveness of anticancer treatment in patients with breast cancer [126,127].

Patients with late-stage (stage III–IV) breast cancer with high PKCλ, c-Met, and ALDH1A3 levels showed a poorer prognosis than patients with low PKCλ, c-Met, and ALDH1A3 levels. Treatment with the c-Met inhibitor foretinib and PKCλ inhibitor auranofin significantly suppressed cell viability and tumor-sphere formation mediated by ALDH1-positive breast CSCs in late-stage basal-like breast cancer. These results suggest that c-Met and PKCλ cooperatively induce poor prognosis in breast cancer [128,129]. Similarly, PKCλ and GLO1 cooperatively promote cell survival in ALDH1-positive breast CSCs, but their inhibition decreases cell viability and tumor-sphere formation [130].

High PKCε levels are associated with shorter disease-free survival in patients with ER-negative breast cancer than in those with ER-positive breast cancer. Although a correlation between PKCε and claudin 1, which is activated by the ERK signaling pathway, was identified in ER-negative cancer, claudin 1 levels are not a prognostic indicator of disease recurrence or survival [131]. Moreover, PKCε-induced activation of TRIM47 stimulates NF-κB signaling, resulting in enhanced breast cancer proliferation and resistance to endocrine therapy [132]. PKCε overexpression in MCF-7 cells increases cell survival by inhibiting apoptosis and inducing autophagy [133]. These results indicate that PKCε is a prognostic marker and therapeutic target in breast cancer.

### 2.5. Colorectal (Colon) Cancer (CRC)

PKCα is involved in cell proliferation, migration, and survival [134] and enhances drug resistance [135] in colon cancer. In colon cancer SW620 cells, PKCα stimulates TF/VIIa/PAR2-induced cell proliferation, migration, and survival through its downstream signaling pathways, ERK1/2/NF-κB [134] and ERK1/2/c-Jun/AP-1 [136]. Mitotic checkpoint kinase Mps1 (also known as TTK) activates the PKCα/ERK1/2 pathway but inhibits the PI3K/Akt pathway, resulting in the promotion of cell proliferation in colon cancer HT-29 and SW480 cells [137].

PKCα activation inhibited DOX-induced apoptosis in HCT15/DOX cells through scavenging of ROS and inhibition of PARP cleavage, whereas siRNA-mediated PKCα knockdown induced apoptosis [135]. Furthermore, PKCα inhibition enhanced resveratrol-induced apoptosis of HT-29 cells [138].

In contrast, the anticancer action of PKCα has been reported in CRC cells. PKCα increased IL12/GM-CSF-mediated M1 polarization of tumor-associated macrophages (TAMs) through the MKK3/6-P38 signaling pathway [139]. Furthermore, PKCα activation inhibited β-catenin-induced transcription and expression of cyclin D1 and c-myc, which are known targets of β-catenin, resulting in the reduced growth of CRC cells [140]. PKCα-deficient *Apc^Min/+^* mice developed a more aggressive histopathological phenotype and had higher mortality than PKCα^+/+^ or PKCα^+/–^ mice [141]. PKCα downregulation is observed at a higher frequency in tissues from advanced CRC stages than in the corresponding normal mucosa [142]. A PKCα mutation was found in CRC samples, but PKCα activation triggered CRC cell death [143]. Interestingly, low PKCα and high Kirsten rat sarcoma viral oncogene homolog (KRAS) expression are associated with a relatively poor prognosis in patients with CRC. PKCα expression in patients decreased in the following order: poorly differentiated < moderately differentiated < well-differentiated adenocarcinoma. However, KRAS levels are correlated with the degree of CRC differentiation [144]. These studies suggest that PKCα may be a potential drug target for CRC treatment. However, further studies are needed to clarify the role of PKCα in CRC cells.

Combined treatment with an atypical PKC inhibitor (ICA-I or ζ-Stat) and thymidylate synthase inhibitor 5-FU synergistically reduced the viability of CRC cells and induced apoptosis and DNA damage [145]. Enhanced PKCζ expression was found in human CRC tissues and cells and correlated with reduced AMPK activation and increased mTOR complex 1 (mTORC1) activation. Silencing of PKCζ inhibited HT-29 cell proliferation via AMPK activation [146]. However, activation of PKCζ inhibited TRAIL-induced apoptosis by regulating survivin levels [147]. Furthermore, PKC-ζ activation increased abnormal growth, proliferation, and migration of metastatic LOVO colon cancer cells via the PKC-ζ/Rac1/Pak1/β-catenin pathway [148]. Phosphorylated PKCζ/λ expression was also higher in colorectal adenocarcinomas than in adenomas. PKCζ/λ overexpression is associated with tumorigenesis in colorectal adenocarcinoma, but PKCζ/λ downregulation is associated with poor prognosis [149]. These results indicate that PKCζ is a useful target for the treatment of CRC.

Dowling’s group reported that PKCβII acts as a tumor suppressor in CRC and that decreased PKCβII level is associated with poor survival outcomes [150]. However, Spindler’s group reported that an increased level of PKCβII is associated with poor prognosis [151]. PKCβI and PKCβII activation increases CRC carcinogenesis and proliferation rates [152]. In COLO205-S cells, PKCβ inhibition increased cell apoptosis through the inactivation of Akt and glycogen synthase kinase-3β (GSK3β) [153].

PKCδ suppresses CRC growth through the activation of p21^Waf1/Cip1^ and p53 [154] but inhibits 5-FU-induced CRC apoptosis [155]. Moreover, PKCδ activation induces CRC cell motility and metastasis via enhanced B7-H4, which plays an important role in cancer growth and immunosuppression. Enhanced expression of PKCδ and B7-H4 is associated with moderate/poor differentiation, lymph node metastasis, and advanced Dukes’ stage [156]. In addition, the activation of PKCδ/NF-κB signaling increases CRC growth, whereas its inhibition results in CRC apoptosis through extrinsic/intrinsic pathways [157]. Increased nuclear translocation of PKCδ in CRC is also associated with worse prognosis [158].

Furthermore, Du’s group suggested that PKCι may serve as a novel therapeutic target for CRC because its inhibition reduces epithelial–mesenchymal transition (EMT), migration, and invasion of CRC cells by suppressing the Rac1-JNK pathway [159]. PKCλ/ι is a key regulator of the interferon pathway. Low PKCλ/ι levels correlate with enhanced interferon signaling and good prognosis in patients with CRC [160].

### 2.6. Gastric (Stomach) Cancer

PKCα is overexpressed in gastric cancer cells and tissues [161,162,163]. PKCα protein overexpression is significantly correlated with age, histologic type, tumor differentiation, depth of invasion, angiolymphatic invasion, pathologic stage, and distant metastasis in gastric cancer [161,162]. Furthermore, PKCα levels were higher in the vincristine-resistant human gastric cancer cell line SGC7901/VCR than in the non-vincristine-resistant cell line SGC7901. PKCα, but not PKCβI, βII, or γ, plays a role in multidrug resistance of SGC7901/VCR cells [163,164]. In HER2-negative advanced gastric cancer, PHD finger protein 8 (PHF8) positively correlates with PKCα, and high PHF8 and PKCα levels are significantly associated with poor clinical outcome [165].

Patients with gastric cancer with high PKCι levels showed lower overall survival compared to those with low PKCι levels [166]. Overexpression of circular RNA of PKCι is positively correlated with poor prognosis in patients with gastric cancer. In vitro experiments revealed that its overexpression promotes proliferation and invasion and reduces apoptosis of gastric cancer cells [167]. Moreover, stathmin 1 expression was significantly associated with gender and poorly differentiated gastric cancer. Furthermore, stathmin 1 expression was significantly correlated with activation-induced cytidine deaminase and PKCι levels [168]. The recurrence of gastric cancer following curative gastrectomy was increased in patients with PKCλ/ι overexpression [169].

These results suggest that PKCα and PKCι may serve as potential prognostic indicators and therapeutic targets for gastric cancer.

### 2.7. Head and Neck Squamous Cell Carcinoma (HNSCC)

HNSCC develops in the mucosal epithelium of the oral cavity, pharynx, and larynx [170], and several PKC isozymes, such as PKCα, β, γ, ε, θ, ι, and ζ are found in HNSCC [171,172,173].

PKCα overexpression occurs more frequently in younger (≤45 years) than older (>45 years) patients with oral tongue SCC (OTSCC). PKCα upregulation is associated with a negative history of alcohol and tobacco consumption. Both overall survival and disease-free survival are impaired in young patients with PKCα overexpression [174]. Furthermore, CC-chemokine receptor 7 and PKCα overexpression in HNSCC are significantly correlated with both cervical lymph node metastasis and clinical stage [175]. Another study also suggested that high PKCα expression is associated with a significantly higher probability of recurrence or death [176].

High levels of autophagy-suppressive circPARD3 are associated with malignant progression and poor prognosis in patients with laryngeal SCC (LSCC). CircPARD3 inhibits autophagy and promotes LSCC cell proliferation, migration, invasion, and chemoresistance through the PKCι/Akt/mTOR pathway [173]. In oral SCC (OSCC), PKCλ/ι expression is positively correlated with malignancy and progression-free survival [177].

High nuclear expression of PKCθ [178] or PKCβII [172] was significantly associated with poor overall survival and rapid recurrence in patients with OSCC, indicating that their nuclear expression can be a potential prognostic marker in patients with OSCC. Furthermore, the expression of CXCR-4, PKCδ, and CD133 is high in poorly differentiated and lymph node metastasis-positive cases of OSCC. CXCR4+/CD133+ and CXCR4+/PKCδ+ double-positive cases show poor survival [179].

### 2.8. Liver Cancer (Hepatocellular Carcinoma)

PKCα levels were higher in biopsy and surgical specimens of hepatocellular carcinoma (HCC) than in adjacent non-cancerous liver tissues [180]. PKCα expression correlated with tumor size and TNM stage. Patients with high PKCα expression showed shorter survival rates than those with low PKCα expression [181]. Inhibition of PKCα expression reduced several migration/invasion-related genes (e.g., *MMP1*, *u-PA*, *u-PAR*, and *FAK*) in both HA22T/VGH and SK-Hep-1 cell lines. Furthermore, PKCα inhibition decreased cyclin D1 levels and increased the levels of p53 and p21^WAF1/CIP1^, resulting in a decreased growth rate of HCC [182]. Enhanced expression of the retinoblastoma protein (RB)-binding transcription factor E2F1 transactivates cell-cycle-related factors and promotes HCC proliferation by activating PKCα [183]. Furthermore, PKCα stimulates dual oxidase 2 (DUOX2)-mediated ROS generation at the post-transcriptional level. DUOX2 inhibition blocked PKCα-induced activation of the Akt/MAPK pathways, as well as HCC cell proliferation, migration, and invasion [184]. A recent study reported that PKCα induces immune evasion and anti-PD1 tolerance by stimulating the zinc finger protein 64/macrophage colony-stimulating factor axis that transforms macrophages to the M2 phenotype to drive immune escape and anti-PD1 tolerance [185].

Suppression of the PKCδ/p38 MAPK pathway induced NF-κB-mediated inhibition of HCC progression [186] and attenuated phosphorylation of heat shock protein 27 that correlates with HCC progression [187]. Blockage of the PKCδ/p38 MAPK/nuclear factor erythroid 2-related factor (Nrf2) pathway also reduced the expression of heme oxygenase-1, which inhibits HCC cell death [188]. In addition, PKCδ triggers HCC progression by increasing mitochondrial ROS generation and HSP60 oxidation and inhibiting RAF kinase inhibitor protein, a negative regulator of MAPK [189]. Hypoxia induces HIF-2α-mediated activation of CUB domain-containing protein 1 (CDCP1) and phosphorylation of PKCδ, which is a downstream factor of CDCP1, leading to stimulation of HCC cell invasion. In fact, CDCP1 expression increases progressively with HCC tumor grade and is negatively correlated with disease-free survival [190]. These studies indicate that PKCδ is a potential prognostic biomarker for HCC.

PKCλ/ι is regarded as a tumor suppressor in HCC. PKCλ/ι levels negatively correlate with HCC histological tumor grade. PKCλ/ι inhibition promotes HCC progression by inducing autophagy, ROS production, and Nrf2 activation [191,192]. Furthermore, PKCβII and PKCθ are downregulated in HCC tissues. Reduced levels of PKCβII and PKCθ are associated with HBV infection and HCC grade, respectively [193]. PKCβ expression was found to be lower in the liver tissues of patients with HCC than in non-tumorous liver tissues [194]. However, another study reported that PKCβ is upregulated in HCC cell lines. Its upregulation increases the migration and invasion of HCC cells [195]. In addition, PKCη expression is downregulated in HCC tissues, and this reduction is associated with poor long-term survival of patients with HCC [196].

### 2.9. Lung Cancer

The two main types of lung cancers are small-cell lung cancer (SCLC) and non-small-cell lung cancer (NSCLC), which are further divided into adenocarcinoma, SCC, and large-cell carcinoma. The roles of PKC isozymes vary in SCLC and NSCLC. For example, following DOX treatment, NSCLC cells showed increased resistance to DOX through PKCα-mediated phosphorylation of Ral-interacting protein (RLIP76) compared to SCLC cells. Depletion of PKCα results in higher growth inhibition in NSCLC cells than in SCLC cells [197]. Phorbol 12-myristate 13-acetate (PMA), also known as 12-O-tetradecanoylphorbol 13-acetate (TPA), induces JNK activation in NSCLC, but not SCLC cells. The absence of JNK activation in PMA-treated SCLC cells was related to the absence of PKCε [198].

PKCα is highly expressed in NSCLCs, and its expression is higher in adenocarcinomas than in SCCs [199]. High PKCα/Rab37/tissue inhibitor of metalloproteinase-1 (TIMP1) expression profile correlated with worse progression-free survival in patients with lung cancer. PKCα-mediated Rab37 phosphorylation stimulated lung cancer cell motility [200]. In lung adenocarcinomas with EGFR mutation, PKCα activation plays a key role in the activation of the Akt/mTORC1 signaling pathway, which is involved in cell survival, growth, and proliferation [201]. PKCα also impairs TRAIL-induced apoptosis in H1299 NSCLC cells by activating the GSK3β/NF-κB pathway, whereas TRIM21 inhibits the activation of NF-κB by GSK3β [202]. Blocking the PKCα/ERK1/2 axis suppresses the proliferation and metastasis of human lung adenocarcinoma A549 cells [203]. In addition, erlotinib-resistant NSCLC cell line H1650-M3 showed substantial upregulation of PKCα and downregulation of PKCδ. Conversely, pharmacological inhibition or RNA interference-mediated depletion of PKCα sensitized H1650-M3 cells to erlotinib [204].

Based on these results, PKCα is regarded as a potential therapeutic target for NSCLC; however, treatment with PKCα-targeted inhibitors has yielded unsatisfactory clinical results [7,205]. Furthermore, a recent study suggested that among the PKC isozymes, high expression of PKCα and the phosphorylation state of PKCα, β, and δ showed the strongest positive correlation with RFS in patients with operable lung adenocarcinomas [206]. Hill’s group also demonstrated that PKCα suppresses KRAS-mediated lung tumor formation by activating the p38 MAPK/TGFβ pathway [207].

In A549 cells, 12-deoxyphorbol esters induce growth arrest and apoptosis by activation of the PKCδ/PKD/ERK pathway [208]. PKCδ activation induced morphological changes and migration of A549 cells by increasing tumor necrosis factor-α (TNF-α)-induced claudin-1 expression [209]. The PKCδ/midkine axis induces hypoxic proliferation and differentiation of A549 cells [210]. Moreover, suppression of the EGFR/PKCδ/NF-κB pathway induced imipramine-triggered anti-NSCLC effects in both in vitro and in vivo models [211]. Interaction of PKCδ with procollagen-lysine,2-oxoglutarate 5-dioxygenase 3 (PLOD3) activates caspase-2 and -4-dependent apoptosis through endoplasmic reticulum stress-induced inositol-requiring enzyme 1α activation and downstream unfolded protein response pathway [212]. Resistance to EGFR TKIs has been observed in EGFR-mutant NSCLC, and nuclear translocation of PKCδ is associated with the response of patients with NSCLC to TKIs. Combined inhibition of PKCδ and EGFR results in a marked regression of resistant NSCLC tumors with EGFR mutations [213]. These results show that PKCδ is involved in cell survival, antiapoptosis, and anticancer drug resistance in NSCLC and thus represents a potential therapeutic target for NSCLC.

Higher expression of PKCε was detected in primary human NSCLC tissue than in the normal lung epithelium [214]. PKCε plays an important role in KRAS-mediated tumorigenesis. Induction of lung tumorigenesis by the carcinogen benzo[a]pyrene, which induces mutations in *KRAS*, was markedly reduced in PKCε-knockout mice [215]. Moreover, PKCɛ is required for NSCLC cell survival and tumor growth. Depletion and inhibition of PKCɛ result in elevated expression of proapoptotic proteins of the Bcl-2 family, caspase recruitment domain-containing proteins, and tumor necrosis factor ligands/receptor superfamily members [216]. Enhanced PKCɛ expression increases XIAP and Bcl-xL levels and anticancer drug resistance in SCLC cells [217]. These results indicate that PKCɛ is an attractive target for lung cancer therapy.

In addition, there was a positive relationship between PKCι expression and c-Myc/GLUT1 signaling in NSCLC. High co-expression of PKCι and GLUT1 is associated with worse prognosis in patients with NSCLC [218]. Poor prognosis and survival in NSCLC are also positively correlated with PKCη expression [219].

Smoking is the most important risk factor for lung cancer. PKCε is involved in smoke-induced activation of tumor necrosis factor-convertase and hyperproliferation of lung cells [220]. High expression of PKCα, β, and δ showed the strongest positive correlation with RFS, depending on the molecular subtype; smoking; and mutational status of *EGFR*, *KRAS*, and *TP53* [206]. In an experiment using the carcinogen nitrosamine 4-(methylnitrosamino)-1-(3-pyridyl)-1-butanone (NNK), which is produced by the nitrosation of nicotine, PKCι activation enhanced the survival and chemoresistance of human lung cancer cells by increasing NNK-induced Bad phosphorylation [221].

### 2.10. Ovarian Cancer

In ovarian cancer, PKCα upregulation is positively correlated with anticancer drug resistance via activation of the PKCα/ERK1/2 or PKCα/JNK signaling pathways [222,223]. Furthermore, increased expression of Wnt family member 5A (Wnt5a) correlates with enhanced metastasis of ovarian cancer via increased vasculogenic capacity, motility, and invasiveness. Wnt5a enhanced vasculogenic mimicry, EMT, migration, and invasiveness of ovarian cancer cells in a PKCα-dependent manner, and inhibition of PKCα blocked these effects [224]. The PKCα/CARMA3 axis plays an important role in the lysophosphatidic acid-induced invasion of ovarian cancer cells [225]. The expression of PKCα, PKCε, and P-gp is higher in epithelial ovarian cancer tissue than in normal, benign, and borderline epithelial ovarian cancer tissues. They were also more highly expressed in the recurrent carcinoma tissues than in patients with initial treatment and were related to poor survival and prognosis in patients with epithelial ovarian cancer [226].

PKCι activation is positively correlated with histopathological grading, International Federation of Gynecology and Obstetrics (FIGO) stage, and poor survival in patients with ovarian cancer [227]. Similarly, Zhang’s group reported PKCι overexpression in most ovarian carcinomas evaluated and a positive correlation between increased PKCι expression and tumor stage or grade [228]. PKCι protein is markedly increased or mislocalized and associated with decreased progression-free survival in epithelial ovarian cancers. In a *Drosophila* in vivo epithelial tissue model, increased PKCι levels resulted in defects in apical-basal polarity, cyclin E expression, and proliferation [229]. siRNA-mediated PKCι silencing led to apoptosis in PKCι-amplified ovarian cancer cells, but not in those without PKCι amplification [230]. The PKCι/angiomotin/Yes-associated protein 1 (YAP1) signaling pathway plays a critical role in ovarian cancer prognosis. PKCι inhibition reduces YAP1 nuclear localization and ovarian cancer growth [231,232]. There was also a positive correlation between PKCι and TNF-α expression. Increased levels of TNF-α and YAP1 promote immune suppression by inhibiting the infiltration of cytotoxic T cells [231]. These results suggest that PKCι may be a therapeutic target and prognostic biomarker for ovarian cancer.

High levels of PKCζ are associated with poor prognosis in human ovarian carcinomas [233]. In ovarian cancer, PKCζ has proapoptotic functions and participates in cell invasion and migration. The PKCζ inhibitor ζ-Stat decreased the invasive behavior of ovarian cancer cells by decreasing the activation of cytosolic Ect2 and Rac1 [234].

### 2.11. Pancreatic, Bile Duct, and Gallbladder Cancer

#### 2.11.1. Pancreatic Cancer

PKCα activation is associated with increased survival, proliferation, migration, and resistance in pancreatic cancer. Hydrophobic motif phosphorylation in PKCα (Ser-657) improves survival in patients with pancreatic adenocarcinoma [235]. Transient receptor potential cation channel subfamily M member 2 (TRPM2) levels were increased in patients with pancreatic ductal adenocarcinoma with increasing tumor stage and showed a negative correlation with overall and progression-free survival time. TRPM2 directly activates PKCα by calcium or indirectly activates PKCε and PKCδ by increasing DAG, leading to activation of the downstream MAPK/MEK pathway [236]. TRAIL-induced apoptosis in pancreatic cancer cells is stimulated by the inhibition of the PKCα/AKT cascade [237]. Chow’s group demonstrated that the TGFβ/PKCα/PTEN pathway is key for the proliferation and metastasis of pancreatic cancer cells [238]. Furthermore, autophagy activation promotes cell survival, proliferation, invasion, and migration in pancreatic cancer [239]. Autophagy activation is dependent on the transcription factor p8, which responds to endoplasmic reticulum stress via the p53/PKCα axis [240]. Several in vivo and in vitro studies suggest that PKCα inhibitors may be of potential therapeutic value against human pancreatic cancers [241,242,243,244]; however, there are no reports of clinical trials using PKCα inhibitors.

High expression of PKCι is associated with poor prognosis in patients with pancreatic cancer [245,246]. High PKCι expression led to increased pancreatic cancer cell growth and migration via the PI3K/AKT and Wnt/β-catenin [246] or Rac1-MEK/ERK1/2 [247] pathways. PKCι is upregulated and activated in pancreatic cancers with mutated KRAS, resulting in increased dephosphorylation and nuclear translocation of YAP1. These changes promote the growth of pancreatic cancer [248]. Inhibition of PKCι alone [249] or in combination with other inhibitors (e.g., specificity protein 1 (Sp1) inhibitor) [250] reduced cell growth and metastasis and induced apoptosis in pancreatic cancer cells. These studies suggest that PKCι can be a promising therapeutic target for pancreatic cancer.

PKCζ activation is positively associated with poor prognosis in patients with pancreatic cancer. It is associated with invasive and metastatic phenotypes of pancreatic adenocarcinoma cells [251]. PKCζ inhibition efficiently reduced pancreatic cancer cell growth and metastasis [249]. PKCζ is a useful immunohistochemical marker for detecting the reverse polarity of invasive micropapillary carcinoma (IMPC) cells. The presence of an IMPC component of <20% was not associated with worse prognosis in patients with pancreatic ductal adenocarcinoma [252].

Enhanced PKCδ expression induces a more malignant phenotype of human ductal pancreatic cancer [253] and is associated with poor survival in patients with pancreatic cancer [254]. Furthermore, PKCδ activation in pancreatic cancer cells increases the expression of MUC1-C oncoprotein, which is associated with the progression of pancreatic cancer [255]. MIST1, a transcription factor, is downregulated in pancreatic cancers [256]. Pancreatic ductal adenocarcinoma showed decreased MIST1 expression, combined with increased nuclear PKCδ accumulation. PKCδ activation increased pancreatic acinar cell dedifferentiation in the absence of MIST1 [257]. Interestingly, following radiotherapy, dying pancreatic cancer cells stimulate the proliferation of living cancer cells via caspase-3/7-dependent PKCδ activation and its downstream Akt/p38 MAPK axis [258]. In addition, PKCδ inhibition may be useful in treating pancreatic cancer with distinct stem-like properties (cancer stem-like cells) [259,260].

PKCθ activation is positively correlated with PKCδ activation and poor survival in patients with pancreatic cancer [254]. In pancreatic cancer cells, MAP4K3 knockdown cells failed to phosphorylate PKCθ, and inhibition of PKCθ activity suppressed insulin-like growth factor-1-mediated cell growth and viability, indicating that the MAP4K3/PKCθ axis may be a therapeutic target for pancreatic cancer [261].

#### 2.11.2. Bile Duct and Gallbladder Cancer

Cholangiocarcinoma (CCA) is a malignant bile duct cancer with a poor prognosis and a low 5-year survival rate (7–20%) [262]. PKCι expression was higher in CCA tissues than in benign bile duct tissues. PKCι expression is positively correlated with cell differentiation and invasion but negatively correlated with E-cadherin expression [263]. PKCι, Snail, and infiltrated immunosuppressive cells are upregulated and associated with poor prognosis in CCA. Although PKCι does not directly interact with Snail, it facilitates EMT and immunosuppression by regulating Snail. PKCι phosphorylates Sp1, and upregulation of phosphorylated Sp1 in CCA tissues is associated with poor prognosis in patients with CCA. Phosphorylated Sp1 regulates Snail expression through the enhanced binding of Sp1 to the Snail promoter [264]. Furthermore, high expression of the adapter protein 14-3-3ζ and PKC-ι was associated with poor prognosis in patients with CCA, and they synergistically induced EMT via the GSK3β/Snail pathway [265]. Therefore, PKCι may be a potential therapeutic target for CCA.

Gallbladder cancer is a rare malignancy with poor prognosis owing to its late diagnosis and rapid progression [266]. PKCι is upregulated and correlates with poor prognosis in patients with gallbladder cancer. PKCι stimulates the aPKCι/Keap1/Nrf2 axis to enhance gallbladder cancer cell growth and drug resistance [267]. Activation of the ASPP2/PKCι/GLI1 cascade promotes cell invasion and metastasis and enhances macrophage recruitment in gallbladder cancer via chemokine ligands (e.g., CCL2 and CCL5) and cytokines (e.g., TNF-α) [268]. Furthermore, PKCϵ was upregulated in peripheral blood samples and stem cells of patients with gallbladder cancer [269]. PKCϵ increased anticancer drug resistance in gallbladder cancer by upregulating MDR1/P-gp [270]. PKCϵ silencing inhibited anticancer drug resistance, proliferation, and colony formation rate and increased apoptosis of gallbladder cancer stem cells [269].

### 2.12. Prostate Cancer

Higher levels of PKCα, β, ε, and η have been detected in malignant prostate tissues than in benign tissues [271]. Moreover, increased PKCα and ζ; decreased PKCβ; and absence of PKCγ, δ, and θ expression were observed in early prostate cancer specimens [272]. However, some studies have reported enhanced PKCδ expression in both low- and high-grade prostate cancer [273,274].

Enhanced PKCα and β activation promotes prostate cancer cell proliferation and growth [275,276], and inhibition of PKCα and β induces apoptosis [276,277]. PKCα activation also increases anticancer drug resistance in prostate cancer cells by increasing Ser70-phosphorylated Bcl-2 and total Bcl-2 protein [278]. In contrast, PKCα activation reduced ATM and increased radiation-mediated apoptosis of androgen-sensitive human prostate cancer cells by stimulating ceramide synthase [279].

PKCδ mediates anticancer drug-induced apoptosis in prostate cancer. For example, apoptosis of prostate cancer cells induced by cystine dimethyl ester [280], PMA [281,282], moracin D [283], and paclitaxel [284] depends on PKCδ activity. PKCδ inhibition represents a potential strategy for treating prostate CSC [116].

PKCε overexpression is positively correlated with prostate cancer development [285,286]. PKCε-mediated signal transducer and activator of transcription-3 (Stat3) Ser727 phosphorylation through integration with the MAPK cascade (RAF-1, MEK1/2, and ERK1/2) is essential for prostate cancer cell invasion [287]. Moreover, PKCε activation has been linked to PTEN loss in prostate tumorigenesis via the CXCL13-CXCR5 pathway [286]. PKCε inhibition also led to significant downregulation of proliferative and metastatic genes, such as *C/EBPβ* (CCAAT/enhancer binding protein β), *CRP* (C-reactive protein), *CMK*, *EGFR*, *CD64*, *Jun B*, and *gp130* [288].

aPKCs (PKCζ and λ/ι) are involved in cell growth, invasion, migration, and apoptosis in prostate cancer. PKCζ expression is positively correlated with poor overall survival in prostate cancer [289]. Inhibitors of PKCζ and λ/ι are therapeutic molecules for prostate cancer [290,291,292,293]. For example, treatment with the aPKC inhibitors 2-acetyl-1,3-cyclopentanedione (ACPD) and ICA-1 significantly decreased malignant cell proliferation and induced apoptosis [290,291]. Furthermore, inhibition of aPKCs attenuates prostate cancer cell metastasis by downregulating vimentin expression [293]. PKCζ inhibition also prevents CXCL12-driven cell migration [292]. Treatment-emergent neuroendocrine prostate cancer (NEPC) is a lethal form of castration-resistant prostate cancer [294]. Interestingly, in NEPC, PKCλ/ι downregulation stimulates serine biosynthesis through the mTORC1/ATF4/PHGDH axis and DNA methylation, resulting in enhanced NEPC differentiation and growth. However, inhibition of DNA methyltransferase activity blocks NEPC differentiation and growth induced by PKCλ/ι downregulation [295]. In addition, two PKCι single nucleotide polymorphisms, rs546950 and rs4955720, are associated with prostate cancer risk in Iranian [296] and Eastern Chinese populations [297]. These results suggest that aPKCs may be potential targets for the prevention and/or treatment of prostate cancer.

### 2.13. Renal Cell Carcinoma (RCC)

RCCs can be classified into four types: clear cells (70–80%), papillary (10–20%), chromophobe (5%), and collecting duct (1%) [298,299]. Expression of PKCα, βI, βII, δ, ε, η, ζ, and ι, but not PKCγ and θ, was observed in patients with clear cell RCC (ccRCC) [300]. Another study reported a relationship between PKCζ, RCC grade, and poor patient survival [301]. Increased PKCη (3 times) and PKCζ (20%) levels were observed in grade 3 and 4 versus grade 1 and 2 ccRCCs [300]. However, PKCα level was decreased in ccRCC versus normal tissue [300,302]. In another study, PKCβI, βII, δ, and ε were expressed in ccRCCs, whereas PKCα, βI, βII, η, and ι were expressed in oncocytoma, a benign kidney tumor [302].

PKCδ activation induces migration of ccRCC cells by stimulating CDCP1 [303] or β1 integrin and FAK [304]. High CDCP1 activation is associated with poor prognosis in patients [303].

PKCε also induces RCC proliferation by regulating β1 integrin [305]. PKCε expression positively correlated with Fuhrman grade and T stage in ccRCC. Inhibition of PKCε activation in the ccRCC cell line 769P inhibited cell growth, migration, and invasion, and it sensitized cells to anticancer drugs by increasing caspase-3 activity [306]. PKCε depletion suppressed the sorting and cancer stem-like phenotype of 769P side population cells by decreasing the ABCB1 transporter and the PI3K/Akt, Stat3, and MAPK/ERK pathways [307]. Moreover, PKCε-mediated claudin-4 phosphorylation induces the EMT phenotype and invasive and metastatic abilities in RCC cells [308]. Another study showed that PKCα and PKCε activation increases the invasive potential of RCC [309]. These results indicate that PKCε may be a potential therapeutic target for RCC.

### 2.14. Skin Cancer

Skin cancers are classified as melanoma and non-melanoma skin cancer (NMSC). The main types of NMSC are basal cell carcinoma (BCC) and SCC [310]. PKCα, δ, ε, ζ, and λ/ι are expressed in melanoma cells [311,312]. PKCβI and βII are expressed exclusively in normal melanocytes or epidermal melanocytes but are downregulated in melanoma cells and benign and malignant melanocytic lesions [311,313,314]. The loss of PKCβ is important for melanoma cell growth [315].

#### 2.14.1. Melanoma

PKCα is overexpressed in melanoma tumor samples and is associated with poor overall survival [316]. PKCα is regarded as a potential therapeutic target for melanoma because it increases melanoma cell invasion by activating the AKT/ERK1/2 axis [317] or, in an αvβ3-dependent manner [318], increases cell proliferation by enhancing the G1 to S transition [319], and it increases melanoma vascularization in a vascular endothelial growth factor receptor-1 (VEGFR1)-independent manner [320].

In melanoma, PKCδ is associated with proapoptotic responses through JNK activation [321] or by inhibition of PKCα/PLD1/AKT signaling [319]. However, another study demonstrated that PKCδ inhibition reduced uveal melanoma cell growth through p53 reactivation [322]. In a recent phase I study, treatment with the PKC inhibitor AEB071 (also known as sotrastaurin) was well tolerated and showed modest clinical activity in patients with metastatic uveal melanoma [323].

PKCζ and ι are also regarded as therapeutic targets for melanoma. In melanoma cells, the aPKC/AKT/NF-κB and PKCι/Par6/RhoA pathways are involved in cell proliferation and increased EMT, respectively. Inhibition of both PKCζ and PKCι reduces EMT and induces apoptosis in melanoma cells [324]. However, PKCι is more involved in melanoma malignancy than PKCζ. Treatment with ICA-1 (PKCι-specific inhibitor) and ζ-Stat (PKCζ-specific inhibitor) reduced melanoma cell proliferation and induced apoptosis, whereas ICA-1 also reduced cell migration and invasion [325].

PKCε-mediated activation of activating transcription factor-2 (ATF2) regulates the migration and invasion of melanoma cells via cellular protein fucosylation. Activated PKCε and ATF2 were observed in advanced-stage melanomas and correlated with decreased cellular protein fucosylation, attenuated cell adhesion, and increased cell motility [326,327]. Furthermore, PKCε is involved in metabotropic glutamate receptor-1-mediated ERK1/2 phosphorylation, resulting in enhanced melanomagenesis and metastasis [328,329].

#### 2.14.2. NMSC

PKCδ plays a protective role in SCC by downregulating p63 and suppressing cell proliferation [330] or by inducing apoptosis in SCC cells [331]. PKCε is involved in ultraviolet radiation (UVR)-induced SCC development. Following UVR treatment, the clonogenicity of isolated keratinocytes increased in PKCε-overexpressing transgenic mice [332]. The PKCε–Stat3 and PKCε–ERK1/2 interactions were also increased in SCC elicited following repeated UVR exposure. PKCε-mediated activation of Stat3 and ERK1/2 increased SCC development [333,334]. In addition, Hedgehog-dependent BCC growth is stimulated by activation of the mTOR/aPKC [335] or aPKC/histone deacetylase axes [336].

### 2.15. Thyroid Carcinoma

The expression of phosphorylated PKCδ along with that of cytokeratin 18, Stat1, HMG-1, p-p70 S6 kinase, Raf-B, glutamine synthetase, and HDAC1 was upregulated in papillary thyroid carcinoma [337]. PKCε expression is reduced in papillary thyroid carcinomas [338]. In anaplastic and follicular thyroid cancer cell lines, PMA treatment stimulates the translocation of PKCα, βI, and δ. PKCδ deletion reduces the PMA-induced antiproliferative effect by inducing cell cycle arrest in the G1/S phase [339]. The expression and localization of PKCβII and PKCδ were observed in medullary thyroid carcinomas. PKCβII inhibition by enzastaurin reduced cell proliferation and survival by inducing caspase-mediated apoptosis and blocking the stimulatory effect of IGF-I on calcitonin secretion [340]. Furthermore, mutated PKCα has been found in pituitary and thyroid tumors [341] and follicular thyroid carcinoma [342,343]. D294G, but not A294G, is a loss-of-function mutation [341,343].

## 3. PKC Isozymes as Diagnostic Biomarkers for Cancer

### 3.1. PKC Isozymes as Diagnostic Immunohistochemical Biomarkers

Compared to normal tissues, overexpression of PKC isozymes in cancer tissues can be used as a diagnostic immunohistochemical biomarker for specific cancer types. For example, higher PKCζ expression was found in invasive ductal carcinoma than in healthy breast tissue [121]. Furthermore, PKCι was significantly upregulated in ovarian cancer compared to normal ovarian tissue. There was a positive correlation between PKCι expression and tumor stage or grade [228]. DOG1 and PKCθ are overexpressed in KIT-negative gastrointestinal stromal tumors, indicating that DOG1 and/or PKCθ may be used in the diagnosis of KIT-negative GISTs [344,345,346]. As mentioned in OTSCC, PKCα was significantly overexpressed in young patients (≤45 years) compared to older patients (>45 years). PKCα overexpression was associated with poor overall and disease-free survival as well as with no alcohol and tobacco consumption. These results indicate that PKCα overexpression may be a novel diagnostic molecular marker for early-onset alcohol- and tobacco-negative high-risk OTSCC [174].

### 3.2. PKC Isozymes as Diagnostic Biomarkers in Body Fluids

Diagnostic cancer biomarkers in body fluids (e.g., blood, urine, feces, or saliva) offer several advantages, such as simple and non-invasive sample collection methods that are less painful in patients, when compared to diagnostic immunohistochemical biomarkers using tissue samples. PKC isozymes are detectable in body fluids as they are secreted by cancer cells [347,348,349].

High levels of activated PKCα have been observed in blood samples collected from cancer-bearing mice [347,348] and patients with lung cancer [350]. However, very low levels of activated PKCα were found in blood samples obtained from healthy mice [347,348] and humans [269]. Furthermore, despite the lack of identification of PKC isozyme, higher serum levels of PKC as well as FAK, MR-1, and Src were identified in patients with AML than in controls [351]. Expression of *PKCε* was significantly reduced in the blood of patients with cervical cancer compared to that in healthy controls [352].

PKCα expression negatively correlates with urinary microRNA (miR)-15a in patients with ccRCC. Increased miR-15a levels were determined in the urine of patients with RCC but were nearly undetectable in oncocytoma, other tumors, and urinary tract inflammation [302]. PKCε downregulation was closely related to miR-31 upregulation [353]. Urinary levels of miR-31 are higher in oncocytomas than in ccRCCs [354]. Recently, our group reported that high levels of activated PKCα were observed in urine samples collected from orthotopic xenograft mice bearing human bladder cancer cells compared with urine samples from normal mice [355]. In urine samples from patients with ccRCC, PKCα levels increased with increasing regression rate. However, PKCι levels were increased in urine samples from patients with oncocytoma but reduced in samples from patients with ccRCC [356].

In addition, increased fecal PKCβII mRNA levels and decreased fecal ζ mRNA levels were found in samples collected from colon cancer-bearing rats compared with those from normal rats [357].

## 4. Summary and Overall Conclusions

PKC isozymes represent potential therapeutic targets in cancer (Table 1). Several natural and synthetic PKC inhibitors have been developed and used in clinical trials. However, most clinical trials using PKC inhibitors with or without other anticancer agents have failed to show significant clinical benefits [7]. Despite these unfavorable results, the fact remains that PKC isozymes constitute attractive therapeutic targets for cancer, and satisfactory clinical results with PKC inhibitors may be obtained when combined with other inhibitors of cancer-related signaling pathways (e.g., TKIs) [7].

Many studies have shown positive relationships between PKC isozymes and poor disease-free survival and survival rates, poor survival following anticancer treatment, and enhanced recurrence (Table 1). Furthermore, several groups have reported differential expression of PKC isozymes by cancer type, for example, PKCθ overexpression in KIT-negative GISTs [344,345,346] or PKCα overexpression in OTSCC [170]. Therefore, PKC isozymes hold great potential as prognostic and diagnostic biomarkers.

PKC-based cancer diagnosis has been performed mainly using tissue samples collected from patients with cancer. Inactivated PKC isozymes are present in the cytosol; however, following activation, PKC isozymes translocate from the cytosol to the inner cell membrane. Several studies have suggested that activated PKC isozymes present in the extracellular space are released into the bloodstream and urine [347,348,349,355]. These studies indicate that PKC isozymes in body fluids (e.g., blood, urine, feces, or saliva) may be potential diagnostic biomarkers for cancer. However, there are very few reports based on PKC isozymes in body fluids. Furthermore, the mechanism by which PKC isozymes are released into bodily fluids remains unclear.

## Figures and Tables

**Figure 1 cancers-14-05425-f001:**
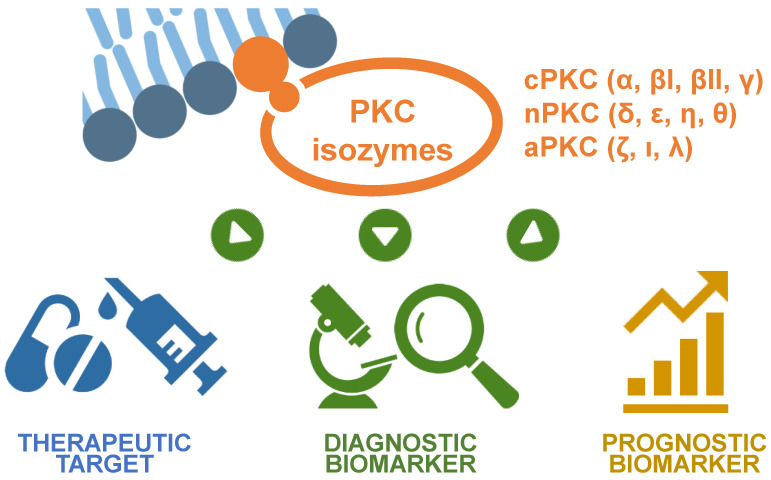
The PKC family consists of at least 11 isozymes that are classified into three subfamilies (cPKC, nPKC, and aPKC). The activation of PKC isozymes is positively associated with poor survival rate, anticancer drug resistance, or increased recurrence in patients with cancer. Furthermore, higher levels of PKC isozymes are found in tissues or body fluids of patients with cancer compared to those in healthy individuals. These data suggest that PKC isozymes represent useful therapeutic targets and potential diagnostic and prognostic biomarkers for cancer.

**Table 1 cancers-14-05425-t001:** PKC isozymes as diagnostic and prognostic biomarkers and therapeutic targets in various cancer types.

Cancer Types	PKC Isozymes	Activity	Effect of Change in PKC Activation on the Cancer	Refs.
Bladder cancer	PKCα	Upregulation	Poor prognosis Increased anticancer drug resistance	[14,15] [16,18]
Blood and bone marrow cancer				
Multiple myeloma	PKCβ	Upregulation	Potential therapeutic target	[24]
Leukemia: lymphocytic leukemia	PKCα	Upregulation	Enhanced chemoresistance	[27,28]
	PKCβ	Upregulation	Potential therapeutic target	[30]
Leukemia: myeloid leukemia	PKCα	Upregulation	Poor survival Promoted anticancer drug resistance	[40] [41,51]
	PKCβ	Upregulation	Enhanced anticancer drug resistance	[52]
	PKCδ	Upregulation	Increased anticancer drug-mediated apoptosis	[45,47]
	PKCε	Upregulation	Poor survival and increased anticancer drug resistance	[43,44]
Myelodysplastic syndromes	PKCα	Upregulation ^(1)^	Induced erythropoiesis	[56]
Lymphoma	PKCβII	Upregulation	Poor prognostic marker and chemotherapeutic target	[63,66]
	PKCδ	Upregulation	Increased anticancer drug-mediated apoptosis	[68,69]
Brain cancer (glioblastoma)	PKCα	Upregulation	Potential therapeutic target Potential prognostic marker	[73,79] [72]
	PKCδ	Upregulation	Antiproliferative and proapoptotic	[90,91]
	PKCε	Upregulation	Potential therapeutic target	[93]
	PKCι	Upregulation	Potential therapeutic target	[83,84]
Breast cancer	PKCα	Upregulation	Poor survival and prognosis Maintenance of migratory and invasive behavior Decreased ER levels and increased antiestrogen resistance Enhanced anti-ErbB-1 sensitivity in ErbB-2-positive breast cancer	[100] [101] [102,103] [111]
	PKCδ	Upregulation	Enhanced mammary tumorigenesis	[114,115]
	PKCθ	Upregulation	Increased migratory and invasive behavior	[122,123]
	PKCε	Upregulation	Decreased disease-free survival	[131]
	PKCη	Upregulation	Enhanced breast cancer malignancy Poor survival following anticancer treatment	[125] [126]
	PKCζ	Upregulation	Increased invasive behavior Poor prognosis, disease-free survival, and survival rate	[121] [120]
	PKCλ	Upregulation	Poor prognosis	[130]
Colorectal (colon) cancer	PKCα ^(2)^	Downregulation Upregulation	Potential therapeutic target Enhanced anticancer drug resistance	[144] [135]
	PKCδ	Upregulation ^(1)^	Increased cancer progression and poor prognosis	[156,158]
	PKCζ	Upregulation	Potential therapeutic target	[145,146]
	PKCι	Upregulation	Potential therapeutic target	[159]
Gastric (stomach) cancer	PKCα	Upregulation	Poor prognosis and increased anticancer drug resistance	[162,164]
	PKCι	Upregulation	Enhanced recurrence of cancer and poor survival	[166,169]
Head and neck squamous cell carcinoma	PKCα	Upregulation	Poor prognosis and survival	[174,175]
	PKCβII	Upregulation ^(1)^	Poor survival and rapid recurrence	[172]
	PKCθ	Upregulation ^(1)^	Poor survival and rapid recurrence	[179]
	PKCι	Upregulation	Increased malignancy and poor survival	[177]
Liver cancer (hepatocellular carcinoma)	PKCα	Upregulation	Poor prognosis and survival Immune escape and anti-PD1 tolerance	[181,184] [185]
	PKCβ	Upregulation	Potential tumor suppressor	[194]
	PKCδ	Upregulation	Potential prognostic marker Poor disease-free survival	[186,189] [190]
	PKCλ/ι	Upregulation	Potential tumor suppressor	[191]
	PKCη	Downregulation	Poor long-term survival	[196]
Lung cancer	PKCα	Upregulation	Potential therapeutic target and poor survival	[200]
	PKCδ	Upregulation	Increased cell survival Increased anticancer drug resistance and potential therapeutic target	[208] [213]
	PKCε	Upregulation	Potential therapeutic target Elevated survival and anticancer drug resistance	[215] [215,217]
	PKCη	Upregulation	Poor prognosis and survival	[219]
	PKCι	Upregulation	Poor prognosis	[218]
Ovarian cancer	PKCα	Upregulation	Poor prognosis and survival Increased anticancer drug resistance	[226] [222,223]
	PKCι	Upregulation	Poor prognosis and survival Potential therapeutic target	[227,228] [230]
	PKCζ	Upregulation	Poor prognosis	[233,234]
Pancreatic, bile duct, and gallbladder cancer				
Pancreatic cancer	PKCα	Upregulation	Potential therapeutic target Potential prognostic marker	[241,244] [236]
	PKCδ	Upregulation	Enhanced cancer progression and poor survival Potential therapeutic target	[254,255] [259,260]
	PKCθ	Upregulation	Poor survival and therapeutic target	[254,261]
	PKCι	Upregulation	Potential prognostic marker and therapeutic target	[246,249]
	PKCζ	Upregulation	Enhanced worse prognosis	[251,252]
Bile duct cancer	PKCι	Upregulation	Potential prognostic marker and therapeutic target	[263,264]
Gallbladder cancer	PKCι	Upregulation	Poor prognosis Enhanced cell growth, migration, and anticancer drug resistance	[267] [268,269]
	PKCε	Upregulation	Enhanced anticancer drug resistance, proliferation, and colony formation rate	[269,270]
Prostate cancer	PKCα	Upregulation	Promoted cell growth and anticancer drug resistance	[275,276]
	PKCδ	Upregulation	Enhanced anticancer drug-induced cell apoptosis	[280,284]
	PKCε	Upregulation	Potential therapeutic target	[288]
	PKCζ	Upregulation	Worse survival and poor overall survival Potential preventive and therapeutic target	[289] [290,292]
	PKCι	Upregulation	Potential preventive and therapeutic target	[290,291]
Renal cell carcinoma	PKCδ	Upregulation	Promoted cancer cell migration	[303]
	PKCε	Upregulation	Potential therapeutic target	[307]
Skin cancer				
Melanoma	PKCα	Upregulation	Poor prognosis and survival Potential therapeutic target for pancreatic cancer stem cells	[316] [317]
	PKCδ	Upregulation	Enhanced proapoptotic response	[319,321]
	PKCζ and ι	Upregulation	Potential therapeutic target	[324]
	PKCε	Upregulation	Potential therapeutic target	[326,327]
Non-melanoma	PKCδ	Upregulation	Protective role in squamous cell carcinomas	[330,331]
	PKCε	Upregulation	Enhanced development of squamous cell carcinomas	[332,334]
Thyroid carcinoma	PKCα	Mutation	Loss of function	[341,343]

^(1)^ Increased nuclear translocation or expression. ^(2)^ Note that there are two different reports, the upregulation or downregulation of PKCα in colorectal cancer.

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
