# Peer review of "Protein Kinase C (PKC) Isozymes as Diagnostic and Prognostic Biomarkers and Therapeutic Targets for Cancer"

_cancers, 2022, doi:10.3390/cancers14215425_

Round 1
Reviewer 1 Report
Comments to the Authors:
The manuscript by Dr. Kawano on “Protein kinase C (PKC) isozymes as diagnostic and prognostic biomarkers, and therapeutic targets for cancer” describes Protein kinase C (PKC) isozymes play key roles in the proliferation, differentiation, survival, migration, invasion, apoptosis, and anticancer drug resistance of cancer cells. Authors highlighted upon therapeutically targeting PKC isozymes for cancer. The article can be modified a little by incorporating few suggestions.
The review is well drafted and designed to discuss the importance of PKC in various types of cancer, and the discussion goes well with the literature cited. Authors need to incorporate minor suggestions to improve the manuscript by including some details about therapeutically targets of PKC in the biomarker section.
Minor concerns:
• Authors need to elaborate the section “3. PKC isozymes as diagnostic biomarkers for cancer” for emphasizing more on how therapeutically target PKC.
• Please undergo a thorough check of the manuscript for typographical and grammatical errors.
Author Response
Reviewer 1
Thank you very much for your kind and valuable comments and suggestions.
Our response to the reviewer’s comments is as follows:
Comment 1) Authors need to elaborate the section “3. PKC isozymes as diagnostic biomarkers for cancer” for emphasizing more on how therapeutically target PKC.
Response) We sincerely appreciate your valuable advice. According to your advice, we have divided the section 3. PKC isozymes as diagnostic biomarkers for cancer into tow subsections, 3.1. PKC isozymes as diagnostic immunohistochemical biomarkers and 3.2. PKC isozymes as diagnostic biomarkers in body fluids. We really want to add several studies to the section 3, but it is a fact that there are very few studies regarding PKC isozymes as diagnostic immunohistochemical biomarkers. However, the following sentences have been added to the section 3.1. Immunohistochemical biomarker:
For example, higher PKCζ expression was found in invasive ductal carcinoma than in healthy breast tissue [121]. Furthermore, PKCι was significantly upregulated in ovarian cancer compared to normal ovary. There was a positive correlation between PKCι expression and tumor stage or grade [228].
Despite these studies, it is a fact that there are few reports regarding PKC isozymes as diagnostic immunohistochemical biomarkers.
Comment 2) Please undergo a thorough check of the manuscript for typographical and grammatical errors
Response) According to your comments, we have carefully checked typographical and grammatical errors (Please see the text).
Reviewer 2 Report
The Manuscript by Kawano et al. is an extensive review on the role of PKCs in cancer. To further improve the paper, the authors could also comment on PKC data in Myelodysplastic Syndromes (MDS), a type of blood cancer, and add some data about PKC involvement in Multiple Myeloma, as recent and less recent papers on the role of PKC in this disease are retrievable.
There are even some minor faults in the text. For instance, authors should indicate Gall bladder Cancer before the abbreviation at Line 612, and they should indicate the complete word before the abbreviation at Line 753 (SCC) and at line 785 (NMSC).
Author Response
Reviewer 2
We are very grateful to you for your valuable and constructive comments.
Our response to the reviewer’s comments is as follows:
Comment 1) The Manuscript by Kawano et al. is an extensive review on the role of PKCs in cancer. To further improve the paper, the authors could also comment on PKC data in Myelodysplastic Syndromes (MDS), a type of blood cancer, and add some data about PKC involvement in Multiple Myeloma, as recent and less recent papers on the role of PKC in this disease are retrievable.
Response) We deeply appreciate your useful comments. We have added a section for myelodysplastic syndromes as follows:
2.2.2.3. Myelodysplastic syndromes (MDS)
MDS are a heterogenous group of hematopoietic stem cell disorders and frequently evolve into AML [54,55]. Nuclear translocation of PKCα induced erythropoiesis in patients with low-risk MDS following treatment with an immunomodulatory drug lenalidomide [56]. Furthermore, the PI-PLCβ1/Cyclin D3/ PKCα signaling pathway was associated with iron-induced oxidative stress and ROS production in MDS patients [57].
<References>
[54] Gangat, N.; Patnaik, M.M.; Tefferi, A. Myelodysplastic syndromes: Contemporary review and how we treat. Am. J. Hematol. 2016, 91, 76‒89.
[55] Cazzola, M. Myelodysplastic syndromes. N. Engl. J. Med. 2020, 383, 1358‒1374.
[56] Poli, A.; Ratti, S.; Finelli, C.; Mongiorgi, S.; Clissa, C.; Lonetti, A.; Cappellini, A.; Catozzi, A.; Barraco, M.; Suh, P.G.; Manzoli, L.; McCubrey, J.A.; Cocco, L.; Follo, M.Y. Nuclear translocation of PKC-α is associated with cell cycle arrest and erythroid differentiation in myelodysplastic syndromes (MDSs). FASEB J. 2018, 32, 681‒692.
[57] Cappellini, A.; Mongiorgi, S.; Finelli, C.; Fazio, A.; Ratti, S.; Marvi, M.V.; Curti, A.; Salvestrini, V.; Pellagatti, A.; Billi, A.M.; Suh, P.G.; McCubrey, J.A.; Boultwood, J.; Manzoli, L.; Cocco, L.; Follo, M.Y. Phospholipase C beta1 (PI-PLCbeta1)/Cyclin D3/protein kinase C (PKC) alpha signaling modulation during iron-induced oxidative stress in myelodysplastic syndromes (MDS). FASEB J. 2020, 34, 15400‒15416.
Comment 2) There are even some minor faults in the text. For instance, authors should indicate Gall bladder Cancer before the abbreviation at Line 612, and they should indicate the complete word before the abbreviation at Line 753 (SCC) and at line 785 (NMSC).
Response) Thank you very much for your kind and helpful comments. To avoid confusion, we have removed the abbreviation of gallbladder cancer in the text. Furthermore, the SCC has been used in 2.7. Head and neck squamous cell carcinoma (HNSCC) (please see Line 453). The NMSC has also been referred in the following sentence:
Skin cancers are classified as melanoma and non-melanoma skin cancer (NMSC) (see Line 757).
In addition, the following sentences have been changed as follows:
In ALL, overexpression of PKCα did not affect cell proliferation, cell cycle, or activation of mitogen-activated protein kinases (MAPKs), but increased chemoresistance through Bcl2 activation [28] (please see Line 128).
PKCα activation is associated with poor survival in patients with AML [40]. PKCα activation also enhanced resistance to chemotherapy in AML cells through Bcl-2 phosphorylation [41], and extracellular-signal-regulated kinase 1/2 (ERK1/2) and Akt activation [42] (please see Line 145).
PKCβII expression in DLBCL was correlated with poor overall and progression-free survival in patients treated with cyclophosphamide, doxorubicin (DOX), vincristine, and prednisolone [63] (please see Line 205).
The two main types of lung cancers are small cell lung cancer (SCLC) and non-small cell lung cancer (NSCLC), which are further divided into adenocarcinoma, SCC, and large-cell carcinoma (please see Line 519).
PKCα is highly expressed in NSCLCs, and its expression is higher in adenocarcinomas than in SCCs [199] (please see Line 529).